# Body Boundary Measurement Using Multiple Line Lasers for a Focused Microwave Thermotherapy System: A Proof-of-Concept Study

**DOI:** 10.3390/s23177438

**Published:** 2023-08-26

**Authors:** Janghoon Jeong, Seong-Ho Son

**Affiliations:** 1Department of ICT Convergence, Soonchunhyang University, Asan 31538, Republic of Korea; 2Department of Mechanical Engineering, Soonchunhyang University, Asan 31538, Republic of Korea

**Keywords:** microwave thermotherapy, boundary measurement, laser vision, image processing

## Abstract

A focused microwave thermotherapy system for non-invasively treating cancerous tumors has recently been actively developed. To accurately focus on the target location, the system needs information about the patient’s body boundary. However, a water bolus is placed between the human body and the microwave applicators to allow the microwave to penetrate the body more easily and cool the body’s skin. The structural configuration makes it difficult to measure the body boundary. In this paper, we present a body boundary measurement method using multiple line lasers and cameras for the application of a focused microwave thermotherapy system. Even with a lack of acquired boundary data, a completely closed boundary line can be reconstructed. In addition, real-time movement tracking is possible as it can be measured quickly, even in situations where the patient is moving, such as breathing and wriggling. The performance is verified with several indicators in a water-filled experimental testbed.

## 1. Introduction

Phased array technology, traditionally used to improve communications and radar performance, is now beginning to be applied to RF and microwave thermal treatment of cancerous tumors [1]. The typical phased array hyperthermia system is presented in [2]. The system applies a microwave frequency of 100 MHz. Recently, there has been a more advanced system called the focused microwave thermotherapy (FMT) system [3,4,5,6]. This system uses microwaves in the 900 MHz band, which are much shorter than the wavelength of the 100 MHz frequency. Therefore, the FMT system can precisely focus microwave energy onto tumors that are a few centimeters or smaller. The focused energy raises the temperature of the cancerous tumor, which results in a therapeutic effect. Because of this, to focus the microwave energy at an accurate location, information on the shape of the patient and the position of the tumor is important.

Until now, a computational study on FMT technology has been conducted [3,4,5,7] and experimentation using some phantoms has been performed [4,6,8]. As can be seen in these studies, information about the position and shape of the body is needed to accurately focus microwave energy on a target within the body. However, it is assumed that the information is already known, but it is difficult to do so in real situations.

The FMT system (see the left side of Figure 1) has a water bolus wrapped around the body that performs the electromagnetic matching of the human body and the cooling of the skin [9]. This environment makes it challenging to measure the shape (boundary line) of the body from the outside.

The microwave tomographic imaging technique attempts to obtain the boundary of a body [5,10]. The technique is based on the electromagnetic inverse problem. Therefore, reconstructing the boundary line is still challenging due to the ill-posedness and convergence instability.

Another approach is an ultrasound–microwave imaging technique [11,12,13]. In the research, boundary information of the breast is first obtained using ultrasound and combined with microwave tomography. This method is not simple to implement because two means, such as ultrasound and microwave, must be used together.

Many studies using the radar-based approach have been conducted to estimate boundaries through experimental verification as well as computational analysis [14,15,16,17]. These include studies that reconstruct boundaries from microwave backscatter measurements to obtain the breast surface data [14,15,16]. For the measurement, a UWB radar sensor was used. To ensure a measurement error within a few millimeters, 32 antennas are required around the breast cross-section layer. Due to installation space and cost issues, when using one sensor, real-time measurement is impossible because it must be scanned mechanically. On the other hand, there is a method for determining distances based on resonance frequency shifts of antennas [17]. This method also requires many antennas to measure the circumferential shape. In particular, since this method uses the shift of the resonant frequency, there is a limitation in that only a distance within one wavelength can be measured.

On the other hand, several studies have been presented to directly measure boundaries by using a point laser system [18,19,20,21,22], especially operating in water or canola oil. However, the measurement time is long (up to 30 min) because the point-by-point measurement is performed with a point laser. Therefore, the method limits its use for tracking the changing boundary of the human body due to respiration. It is an important issue in the FMT that microwave energy should be focused only on a desired tumor location to increase the efficiency of tumor treatment [23,24,25,26].

In this paper, we present a line laser vision-based method for measuring the cross-sectional boundary line of a human body surrounded by a water bolus. A major advantage of line lasers compared to point lasers is their ability to capture multiple points simultaneously [27]. This offers the possibility to measure even a small number of lasers in real time. On the other hand, there is a challenge of correcting the distorted camera image generated by the refraction of light by the water bolus. As a disadvantage, there is a limit to its use because the laser beam cannot penetrate the opaque bolus. Furthermore, we overcome situations where measurement data are insufficient due to the limited field-of-view of the used line laser. It also demonstrates real-time measurement performance in situations where the body is moving.

The remainder of this paper is organized as follows. In Section 2, a proposed boundary measurement method and the experimental testbed system are presented. In Section 3, some tests and results are shown, including the case of insufficient measurement data. Finally, we summarize and conclude in Section 4.

## 2. Materials and Methods

In this section, we present a proposed measurement system and data processing method to obtain the cross-section boundary of the body. In particular, we describe a method for reconstructing a closed boundary line even in the case of insufficient measurement data; for example, one that is caused by a limited field-of-view of used line laser or the lack of number of line lasers. We also consider dynamic situations, such as breathing and wriggling of the body. Lastly, we present several indicators to evaluate measurement performance.

### 2.1. Measurement System and Preprocessing

Considering a typical FMT system, we propose a method that measures the cross-section boundary line from the outside of a body surrounded by a water bolus, as shown on the left in Figure 1. The experimental testbed system for proof-of-concept is shown on the right in Figure 1. The testbed mainly consists of line lasers and cameras that acquire boundary images from four body sides. To control them and send the camera image to the main computer, a microcontroller (Raspberry Pi) is used.

Each set of the line laser and camera is mounted on the outside of the tank to illuminate the cross-section of the body. The tank is filled with water to simulate a water bolus surrounding the body. The camera image is transmitted wirelessly to the main computer, and the computer performs an algorithm to extract the body boundary.

To obtain the boundary line, only the red strip of line laser illuminated onto the body is first extracted from the camera image (see Figure 2a,b). The main logic is applied as follows.
(1)$$\begin{array}{cc} \mathrm{if}\;((I_{R} & \left(i,\;j)-I_{G}(i,\;j)\geq threshold)\;\mathrm{and}\;(I_{R}\left(i,\;j\right)-I_{B}\left(i,\;j\right)\geq threshold)\right)\\ & \left\{\begin{array}{c} I_{R}\left(i,\;j\right)=I_{R}\left(i,\;j\right)\\ I_{G}\left(i,\;j\right)=I_{G}\left(i,\;j\right)\\ I_{B}\left(i,\;j\right)=I_{B}\left(i,\;j\right) \end{array}\right.\\ \mathrm{else}\\ & \left\{\begin{array}{c} I_{R}\left(i,\;j\right)=0\\ I_{G}\left(i,\;j\right)=0\\ I_{B}\left(i,\;j\right)=0 \end{array}\right. \end{array}$$where $I_{R}$, $I_{G}$, and $I_{B}$ are each color matrix of the RGB image and $\left(i,\;j\right)$ is the index of the matrix component (i.e., pixel). The matrix size depends on the image resolution of the camera (1280 × 720 in this work). It is helpful to lower the resolution for faster calculation by reducing the amount of calculation. However, it should be noted that measurement errors can be large.

The width of the extracting red stripline depends on the set threshold. The width affects the performance and computation time of binarization (Figure 2c) and thinning (Figure 2d). The thinning process is necessary to extract the centerline of the laser stripline. This is because the non-uniform stripline width is generated due to the illumination performance of the line laser or the non-uniformity of the surface of the body. In this work, binarization is performed using Otsu’s algorithm for automatic image thresholding [28], and thinning is applied using a universal algorithm for image skeletonization [29].

In this way, preprocessing is performed from the originally captured image (Figure 2a) to the extraction of laser centerline pixel data (Figure 2e).

### 2.2. Converting Pixel Data to Real-World Coordinates

Now, we need to convert the pixel coordinates (Figure 2e) to the real-world coordinates (Figure 2f). To obtain that conversion relation through experimentation, a black and white checkerboard is used. In the water tank, the board is placed on the same plane defined by the line laser (see Figure 3a). The grid size of a checkerboard is 10 mm × 10 mm. The checkerboard is captured by a camera mounted on the outside of the tank. Then, we select grid points on the image and collect the pixel coordinate data. Figure 3b shows the grid points in the pixel coordinate system. Meanwhile, we know that the spacing between the grid points is 10 mm. Therefore, the world coordinates corresponding to each grid point can be prepared, as shown in Figure 3c.

The conversion relation can be set to functions $x=f\left(u,v\right)$ and $y=g\left(u,v\right)$, where $\left(u,v\right)$ is pixel coordinates and $\left(x,y\right)$ is the world coordinates. To obtain the function, we use a biharmonic spline interpolation method based on Green’s function [30,31]. The biharmonic equation can be expressed as [31]:(2)$$S\left(x\right)=T\left(p\right)+\sum_{j=1}^{N}w_{j}G\left(p,p_{j}\right)\;$$
where x is the *x*-coordinate value in the world coordinate system and $G\left(p,p_{j}\right)$ is Green’s function. p is the arbitrary pixel point, $p_{j}$ is the *j*-th pixel point (grid point) in pixel coordinates, and $w_{j}$ is the associated unknown weights relative to p. Here, $T\left(p\right)$ is a trend function that captures variations that may not be expressible via Green’s functions or a regional trend. Thus, this function may need to be removed first from the data constraints [31]. The weight $w_{j}$ is decided by requiring Equation (2) to be satisfied exactly at N pixel data locations, i.e.:(3)$$s\left(x_{i}\right)=\sum_{j=1}^{N}w_{j}G\left(p_{i},p_{j}\right)$$
where $x_{i}$ is the *x*-coordinate value in a known world coordinate system for converting pixel coordinates. A 2D Green’s function $G\left(p_{i},p_{j}\right)$ for biharmonic spline interpolation can be computed as follows [30].
(4)$$G\left(p_{i},p_{j}\right)=||p_{i},p_{j}|{|}_{2}\;\ln(||p_{i},p_{j}|{|}_{2}-1)$$

The function $y=g\left(u,v\right)$ can also be obtained in the same way using Equations (2)–(4). In addition, each conversion function can be obtained for four cameras. On the other hand, the boundary lines obtained from the four cameras can then be merged to reveal a fully closed section boundary of the body (see Figure 2g).

### 2.3. Full Boundary Reconstruction with Insufficient Data

As the number of line lasers used in the measurement system increases, the complexity and cost of the system also increase. Therefore, it is desirable to use as few line lasers as possible. On the other hand, if the number of line lasers and cameras is reduced, the laser lines may not cover the entire circumference of the body to be measured (see Figure 4a).

The length of the laser line projected onto the body surface depends on the field-of-view of the line laser used, i.e., the illumination angle and the distance between the body and the laser. In this work, the illumination angle of the line laser used is 90 degrees, and the distance between the laser and the body to be measured is about 100 mm. This condition considers that the body boundary measurement module should be compactly applied to the practical FMT system. However, considering the patient’s breathing, the distance can be as close as 100 mm to 70 mm. In this case, the line laser illuminated on the body is calculated as about 140 mm.

According to [32], the waist circumference of Asians with abdominal obesity is about 900 mm. Therefore, at least seven line lasers are required to illuminate the full boundary line. However, in our measurement system, we only use four line lasers (see Figure 1). Under these conditions, measurement data may be significantly lacking, as shown in Figure 4a. In this work, we use a smoothing spline-based interpolation technique to fill the gaps.

Direct interpolation in the form of a closed curve in Cartesian coordinates is difficult. Therefore, we first represent the measured data in a polar coordinate system (see Figure 4b) and then apply a smoothing spline-based interpolation to reconstruct a fully closed curve (cross-sectional boundary of the body). The smoothing spline can be created using nonparametric regression by minimizing the function as follows.
(5)$$f\left(x\right)=\arg\min(p\sum_{i=1}^{N}{\left(y_{i}-g\left(x_{i}\right)\right)}^{2}+\left(1-p\right)\int_{-\pi }^{\pi }\frac{d^{2}g\left(t\right)}{dt^{2}}dt)$$
where N is the number of data and −π and π are the start and stop angles of the spline. p is a hyperparameter that determines the strength of regularization and is defined between 0 and 1. If p is small, the curve will be smoother, whereas a large p produces a wavier curve that attempts to fit each of the individual data. The hyperparameter is determined automatically using the generalized cross-validation (GCV) method [33]. After interpolation (see Figure 4c), the data move back into the Cartesian coordinate system (see Figure 4d). In this way, the entire section boundary in the form of a closed curve can be obtained.

### 2.4. Consideration of Body Movement

To accurately focus the microwave energy on the location of the tumor in the body, it is necessary to track the movement of the patient’s breathing and wriggling in real time.

According to [34,35], the average respiratory rate of a healthy adult is 14–20 breaths per minute. This means that each breath takes about 3.0–4.3 s. We can consider sampling 10× faster to obtain a good measurement of the signal without aliasing. Then, the sampling time will be 0.3–0.43 s (i.e., 2.3–3.3 Hz). Ultimately, monitoring a patient’s breathing requires a sampling rate of at least 2 Hz or higher.

The images acquired from the camera go through the image and data processing and data conversion processes described in Section 2.1 and Section 2.2, and the reconstructed boundary is monitored on the screen through the boundary reconstruction process described in Section 2.3. This process should be repeated at least 2 Hz (i.e., within 0.5 s), making it possible to monitor the boundary of the moving body. We evaluate whether our measurement system is capable of this. It is implemented as a MATLAB program running on a computer with a 3.20 GHz Intel Core i9-12900k.

### 2.5. Performance Indicators

To evaluate the measurement performance, we test two body models made on a 3D printer using a white PLA filament. One is a circular model with a diameter of 180 mm and the other is an elliptical model with major and minor axes of 200 mm and 120 mm, respectively. Strictly speaking, the patient’s body shape is irregular. However, most cross-sectional shapes fall into the category of circular or elliptical. Therefore, we used these two regular models to evaluate the accuracy of our proposed body boundary measurement system. In the future, it is necessary to verify it with the actual human body.

To simulate insufficient measurement data, black masking tape is attached along the cross-sectional line to be measured. The laser is then absorbed by the tape and is not visible to the camera. In this study, a boundary measurement test is conducted with a model where 30%, 50%, and 70% of the entire circumference were attached with black tape. The fully reconstructed boundary is then evaluated through the difference from the original shape. On the other hand, both measured and exact boundary lines are closed curves. Therefore, we convert the data to polar coordinates for easy technical comparison and compute the distance errors over the angle. Based on the errors, we evaluate the measurement performance with several indicators.

Simply, we can use an indicator, such as peak absolute error (PAE).
(6)$$\mathrm{PAE}=max_{i=1}^{N}\left|r_{i}-{\hat{r}}_{i}\right|\;$$
where $r_{i}$ and ${\hat{r}}_{i}$ are the distances calculated in polar coordinates of the exact boundary and measured (or reconstructed) boundary, respectively. However, it should be noted that PAE may contain unexpected outliers in the data.

For outliers to be insensitive, mean absolute error (MAE) and root-mean-square error (RMSE) can be used as indicators for measuring performance, as follows.
(7)$$\mathrm{MAE}=\frac{1}{N}\sum_{i=1}^{N}\left|r_{i}-{\hat{r}}_{i}\right|\;$$
(8)$$\mathrm{RMSE}=\sqrt{\frac{1}{N}\sum_{i=1}^{N}{\left(r_{i}-{\hat{r}}_{i}\right)}^{2}}$$

MAE is an evaluation indicator that averages the size of errors, and lower values are better. In cases where the errors in the data from the two compared groups come out as positive and negative, they can cancel out each other, causing distortion. However, since MAE expresses the error as an absolute value, it can prevent distortion. RMSE is the value obtained by taking the square root of the mean square error (MSE), and it can prevent distortion compared to the sensitive MSE to outliers. Like MAE, the lower the RMSE, the closer the measured boundary is to the exact boundary. It is also an indicator of how far it deviates from the center of the exact boundary. Since the unit of the RMSE is mm, it is easy to understand intuitively. However, there is still debate over which metric is better [36].

As another indicator, we can use the similarity rate (SR) between the measured and exact boundaries, defined as follows.
(9)$$\mathrm{SR}=\left(1-\frac{1}{N}\sum_{i=1}^{N}\left|\frac{r_{i}-{\hat{r}}_{i}}{r_{i}}\right|\right)\times 100$$It is determined by summing the ratios of distance differences between the exact and measured boundaries for each angle. This means that the closer the value is to 100, the smaller the error.

## 3. Results and Discussion

In this section, we show the results of the proposed boundary measurement method and discuss them. This includes reconstructing a full cross-sectional boundary line with insufficient measurement data and demonstrating the real-time measurement performance in the presence of body motion.

### 3.1. Measurement of Boundary Line from Line Laser

According to the method described in Section 2.1 and Section 2.2, we conducted a test to obtain a merged boundary line from multiple line lasers. As an example, Figure 5a shows the test setup for a circular test model (cylinder type) with a diameter of 180 mm. Another test model is an elliptical cylinder with major and minor axes of 200 mm and 120 mm, respectively.

The measurement results for the two test models are shown in Figure 6a,c and are compared to exact boundary data. The error shows approximately +/−1.5 mm for the circular model (Figure 6b) and +/−2.5 mm for the elliptical model (Figure 6d). To quantitatively evaluate the accuracy of the obtained boundary line, we used the performance indicators presented in Section 2.5. The results are summarized in Table 1.

For the circular test model with a diameter of 18 mm, the peak absolute error, PAE, is 1.67 mm. The mean absolute error, MAE, and the root-mean-square error, RMSE, are 0.60 mm and 0.73 m, respectively. The similarity rate is 99.33%. On the other hand, the measurement performance of the elliptical model is slightly worse than the circular one, but the difference is not significant.

As a result, the measurement performance is excellent in all indicators. This means that the measured boundary line closely matches the actual boundary line.

### 3.2. Full Boundary Reconstruction with Insufficient Data

In this section, we show the full boundary reconstruction performance for a case where the measurement data are lacking because the line lasers used do not fully illuminate the body circumference. For the reconstruction, the method described in Section 2.3 was applied. To implement the lack of measurement data, a certain ratio of the measurement section was attached with black masking tape. In this study, cases of 30%, 50%, and 70% loss were considered. Among them, Figure 7 shows the reconstruction result (blue dashed line) for the case of 70% loss across the entire section boundary line. Importantly, although a large part of the data were lost, the four orthogonal boundaries (red solid lines) that characterize the shape of the circumference were measured during actual measurement with the cameras.

Table 2 shows the boundary reconstruction performance according to measurement data loss (30%, 50%, and 70%). It is also informative to compare these results with performance when full measurement data are available (i.e., 0% data loss, see Table 1). It shows that the reconstruction performance with up to 50% data loss is almost the same as no data loss. That is, for both the circular and elliptical test models, the PAE is within 2 mm, the MAE and RMSE are within 1 mm, and the SR is over 99%. On the other hand, in the case of 70% data loss, the error slightly increased, but this is negligible in FMT applications. As such, even with 70% data loss, the complete boundary was reconstructed well. It is expected that this is because the data of the four parts (top, bottom, left, and right) corresponding to the principal axes that determine the morphological characteristics were used the least.

### 3.3. Measurement Time

As explained in Section 2.4, it is important to measure the boundary of the body even in situations where there is motion, such as breathing and wriggling. This is because microwave energy can be focused while following the target (tumor) to be focused. In this section, we evaluate measurement performance in the presence of body motion.

For the experiment, we gently moved the body model by hand in the measurement testbed filled with water (see Figure 5a,b). As a result, Figure 8 shows the measured boundary line every second moving clockwise with the exact boundary line from the 11 o’clock position for a circular model with a diameter of 180 mm. It can be seen that the measured boundary is almost as circular as the original shape of the test model.

To evaluate the real-time measurement performance, we summarized the measurement error every second. Table 3 shows the boundary measurement performance every second. Compared to the static situation, the overall evaluation performance of the moving model shows a decrease. The cause lies in the fact that as the model moves, errors due to external factors, such as water turbulence, and factors related to processing time occur when measuring the boundary line. In some cases, a PAE of 3 mm or more occurred, but when measuring the cross-sectional boundary of the patient, the overall evaluation performance was satisfactory. The SR was over 98%. The variation during a patient’s breath is expected to be smaller than the movement of the model. When applied to actual patients, it is anticipated that with accurate boundary estimation, breath monitoring could be feasible. In conclusion, no significant difference was found in these movement test results compared to the results measured in a static state.

On the other hand, we investigate the processing time for the measurement. The result is summarized in Table 4. According to the cumulative time, one cycle takes 428 ms (i.e., 2.3 Hz). This performance satisfies the minimum requirement of 430 ms mentioned in Section 2.4. Here, one cycle includes the steps from taking the line laser with a camera to generating the final boundary line and outputting it to the monitor. In particular, the largest amount of time is occupied in the image processing step. In this work, we used the MATLAB program to implement the proposed measurement algorithm. However, when implemented in a compilation-based language (such as C/C++), the processing time can be drastically reduced.

## 4. Conclusions

In this paper, we presented a body boundary measurement method applicable to an FMT system operating in a water bolus environment. The method is implemented with multiple line lasers and camera image processing techniques. For the proof-of-concept study, we developed an experimental measurement system consisting of four sets of line lasers and cameras. It was installed in four places that can characterize the overall cross-sectional shape of the body to be measured. Through this, we confirmed that the entire boundary was well-reconstructed with only 30% of the data. The similarity rate is over 98%. On the other hand, the body boundary line can be created every 428 ms in the developed experimental testbed. This is the extent to which the patient’s breathing and wriggling can be followed. If the algorithm is implemented in the C/C++ language, much faster processing will be possible. In the next study, it will be necessary to verify the boundary measurement performance with actual human subjects. These considerations will eventually help develop a practical FMT system.

## Figures and Tables

**Figure 1 sensors-23-07438-f001:**
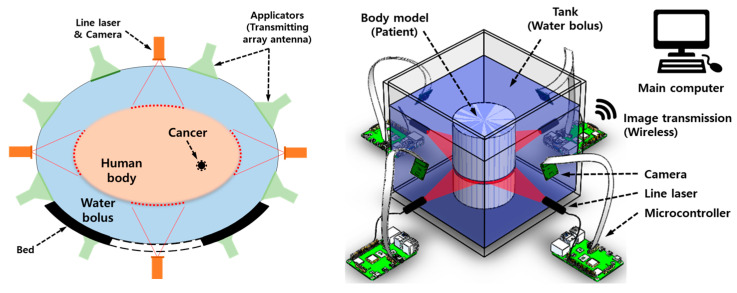
Cross-sectional configuration of a focused microwave thermotherapy system (**left**) and an experimental testbed for the proposed body boundary measurement (**right**).

**Figure 2 sensors-23-07438-f002:**
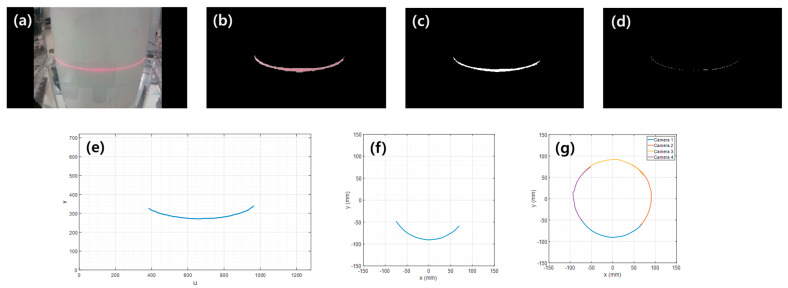
Procedure for section boundary measurement: (**a**) image capture with a camera, (**b**) laser stripline extraction, (**c**) binarization, (**d**) laser centerline extraction, (**e**) pixel data of laser centerline, (**f**) converting pixel data to real-world coordinates, (**g**) merging boundary lines obtained from four directions.

**Figure 3 sensors-23-07438-f003:**
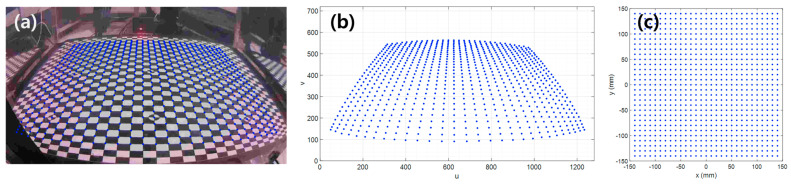
Converting a camera image to real-world coordinates: (**a**) captured checkerboard image and selected grid points, (**b**) pixel coordinate data of selected grid points, (**c**) real-world coordinate data corresponding to each pixel coordinate data.

**Figure 4 sensors-23-07438-f004:**
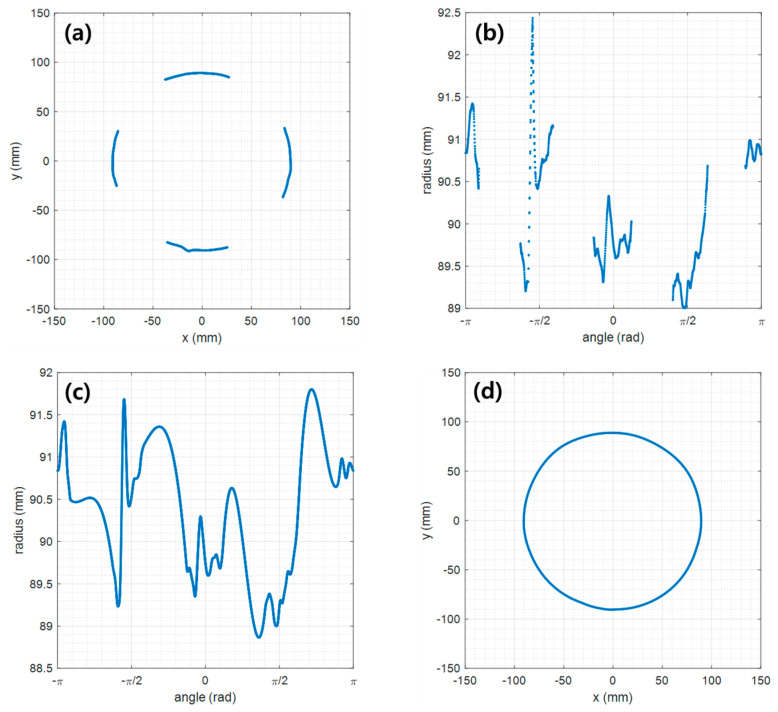
Procedure for full boundary reconstruction with insufficient measurement data: (**a**) incomplete boundary line with gaps, (**b**) converting to polar coordinates, (**c**) interpolating to fill gaps, (**d**) reconverting to Cartesian coordinates.

**Figure 5 sensors-23-07438-f005:**
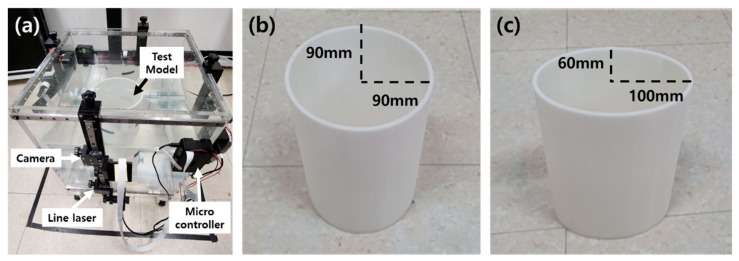
Experimental setup: (**a**) boundary measurement testbed, (**b**) circular test model, (**c**) elliptical test model.

**Figure 6 sensors-23-07438-f006:**
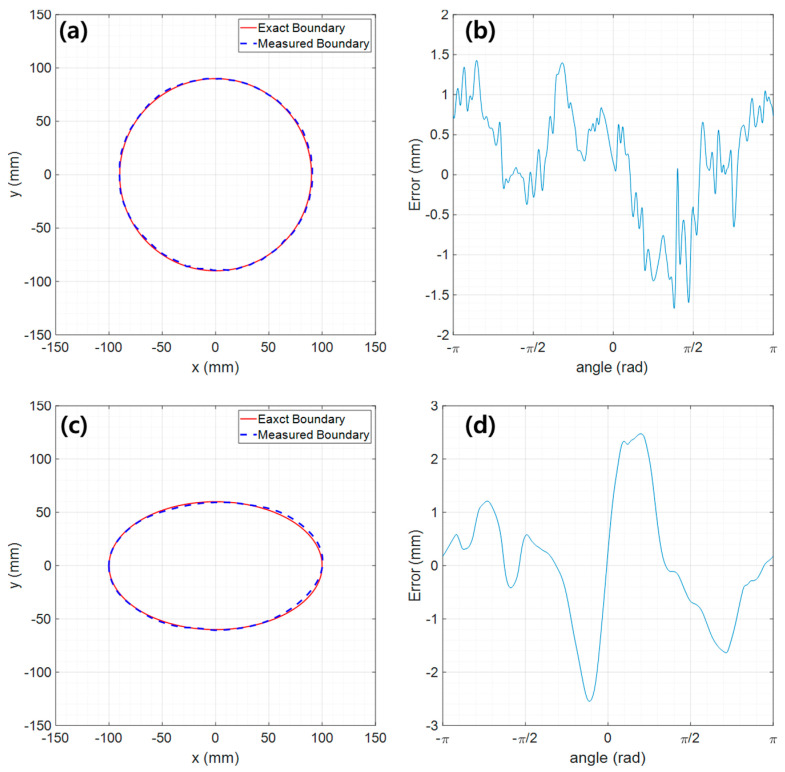
Boundary measurement results: (**a**) boundary line of the circular model, (**b**) measurement error of the circular model, (**c**) boundary line of the elliptical model, (**d**) measurement error of the elliptical model.

**Figure 7 sensors-23-07438-f007:**
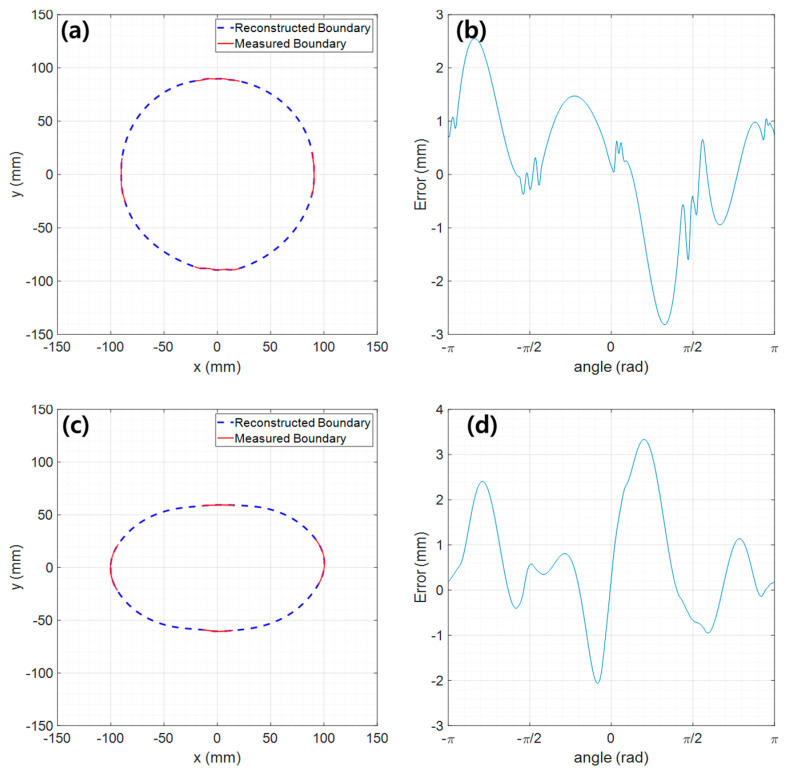
Results of full boundary reconstruction under conditions of lack of measurement data: (**a**) boundary line of the circular model, (**b**) reconstruction error of the circular model, (**c**) boundary line of the elliptical model, (**d**) reconstruction error of the elliptical model.

**Figure 8 sensors-23-07438-f008:**
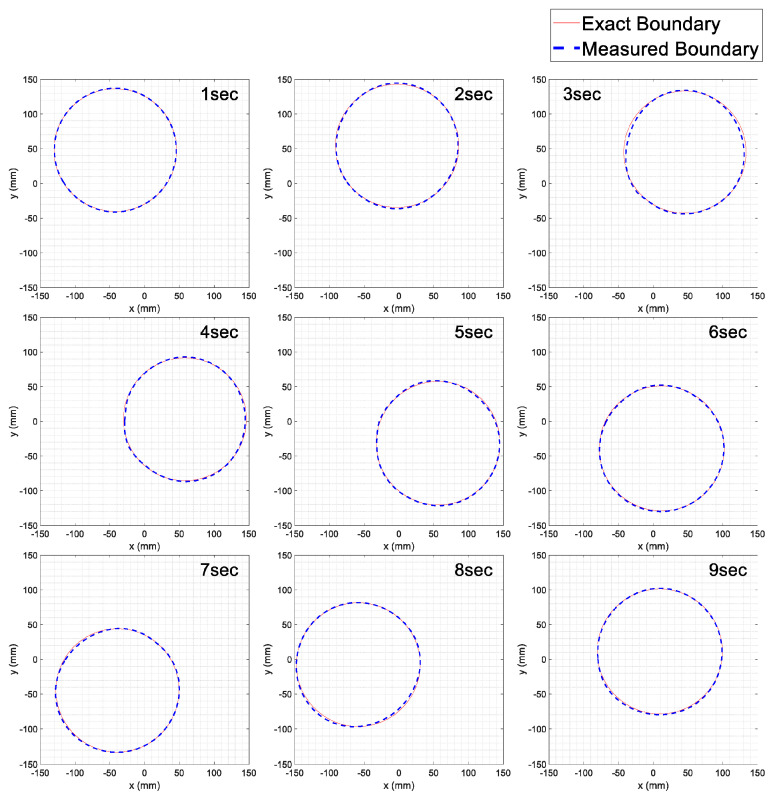
Real-time measurement results of the entire section boundary for a circular model with a diameter of 180 mm.

**Table 1 sensors-23-07438-t001:** Boundary measurement performance using various indicators.

Test Model	PAE (mm)	MAE (mm)	RMSE (mm)	SR (%)
Circle	1.67	0.60	0.73	99.33
Ellipse	2.55	0.90	1.17	98.83

**Table 2 sensors-23-07438-t002:** Boundary reconstruction performance according to measurement data loss.

Test Model	Data Loss (%)	PAE (mm)	MAE (mm)	RMSE (mm)	SR (%)
Circle	30	1.67	0.53	0.66	99.42
	50	1.67	0.67	0.78	99.26
	70	2.81	1.03	1.27	98.86
Ellipse	30	2.55	0.84	1.11	98.82
	50	2.55	0.87	1.13	98.88
	70	3.34	1.03	1.36	98.66

**Table 3 sensors-23-07438-t003:** Boundary measurement performance in real-time experiments.

Time (s)	PAE (mm)	MAE (mm)	RMSE (mm)	SR (%)
1	0.84	0.45	0.51	99.49
2	1.54	0.92	1.03	98.97
3	3.25	1.60	1.79	98.16
4	2.23	0.90	1.05	98.98
5	2.04	0.78	0.94	99.13
6	1.25	0.74	0.81	99.18
7	2.37	0.75	1.00	99.15
8	2.63	0.95	1.30	98.93
9	1.81	0.84	0.95	99.06
Average	2.00	0.88	1.04	99.01

**Table 4 sensors-23-07438-t004:** Processing time for full boundary measurement.

	Image Acquisition	Image Processing	Data Calibration and Merging	Reconstruction of Full Boundary	Display on Monitor
Step time (ms)	72	286	30	20	20
Cumulative time (ms)	72	358	388	408	428

## Data Availability

Not applicable.

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
