# Peer review of "Body Boundary Measurement Using Multiple Line Lasers for a Focused Microwave Thermotherapy System: A Proof-of-Concept Study"

_sensors, 2023, doi:10.3390/s23177438_

Round 1

Reviewer 1 Report

This manuscript described a method to determine the boundary of patient in the microwave thermotherapy system. Such method can fast delineate the boundary of object, and shows potential clinical application. Some minor issues are need to be addressed. 

1. The authors only tested the regularly shaped object such as circles and ellipses. How about objects with irregularly shapes? In practice, the patient's body shape is irregular.

2. The authors only discussed the effect and tracking error of their own system. How about its effect compared with other similar systems?

3. In practical applications, cost is a problem that must be considered. Does the system developed by the author have certain advantages in terms of cost?

Reviewer 2 Report

1. The manuscript provides an overview of a focused microwave thermotherapy system for non-invasively treating cancerous tumors, which is interesting and relevant. However, it lacks essential technical details about the system's configuration in the abstract, such as the frequency used, power levels, and the specific targeting mechanism employed. Including this information in the abstract would enhance the understanding of the system's capabilities and potential applications for cancer treatment.

2. While the paper highlights the significance of accurate body boundary information for the focused microwave thermotherapy system, it lacks sufficient clarity regarding the proposed body boundary measurement method using multiple line lasers. Details regarding the accuracy, precision, and potential limitations of this method are crucial for evaluating its efficacy. Further elaboration on how the reconstructed boundary line is verified and compared with actual boundary data in the water-filled experimental testbed would strengthen the paper's findings.

3. The literature review provides a comprehensive overview of various boundary measurement techniques for focused microwave thermotherapy (FMT) systems. However, it would be beneficial to include a critical analysis of the strengths and limitations of each approach. 

4. Additionally, the review could further emphasize the existing gaps in the research and propose potential areas for future investigation to address the challenges associated with accurately measuring the body boundary in real-time, especially when the patient is in motion.

5. While the literature review extensively covers different boundary measurement methods for FMT systems, it could benefit from providing more specific details about the advantages and disadvantages of the line laser vision-based method being presented in the paper. 

6. The review should also elaborate on how this new approach differs from previous techniques and highlight its potential impact on improving the accuracy and efficiency of focused microwave thermotherapy. 

7. Including comparative analysis with existing methods would enhance the understanding of the novelty and significance of the proposed line laser vision-based approach.

8. The study presents an interesting method for obtaining a merged boundary line from multiple line lasers for circular and elliptical test models. The results show promising accuracy, with a peak absolute error of 1.67 mm for the circular model and slightly higher, but still acceptable, errors for the elliptical model. The quantitative evaluation and performance indicators presented in Table 1 provide valuable insights into the measurement accuracy.

9. However, it would be beneficial to include more details on the statistical significance of the differences between the measurement performances of the circular and elliptical models. Additionally, the paper could provide a discussion on potential factors contributing to the observed differences in accuracy between the two models.

10. In Section 3.2, the authors demonstrate the ability to reconstruct the full boundary even with a significant lack of measurement data (70% loss). This is a noteworthy finding that showcases the robustness of the proposed method. To further enhance the paper, it would be helpful to provide additional information on how this performance was achieved, such as the methodology and algorithms used for boundary reconstruction in the absence of complete data.

11. The study also evaluates the processing time for the measurement, which is essential in practical applications. The performance meets the minimum requirement mentioned in Section 2.4. However, to enhance the applicability of the proposed method, it would be beneficial to provide insights into potential avenues for optimizing the image processing step, particularly in implementation using a compilation-based language like C/C++.

12. Authors have presented the Real-time measurement results of the entire section boundary for a circular model, but in real application it’s not circular, will this affect the performance of the system? And if yes, how you will overcome those challenges? 

Round 2

Reviewer 2 Report

They have responded to all comments.